# An Evaluation of Approaches to
# Train Embeddings for Logical Inference

## Yasir White[1], Jevon Lipsey[2], Jeff Heflin[3]

[1]Los Angeles Pierce College
[2]Colorado College
[3]Lehigh University
y.white005@gmail.com, jevonlipsey1029@gmail.com, heflin@cse.lehigh.edu

## Abstract

Knowledge bases traditionally require manual optimization to ensure reasonable performance when answering queries. We build on previous neurosymbolic approaches by improving the training of an embedding model for logical statements that maximizes similarity between unifying atoms and minimizes similarity of non-unifying atoms. In particular, we evaluate three approaches to training this model by increasing the occurrence of atoms with repeated terms, mutating anchor atoms to create positive and negative examples for use in triplet loss, and training with the "hardest" examples.

## Introduction

First-order logic is a powerful system for describing logical theories that has been used to formalize various mathematical concepts, such as number theory and set theory. Various algorithms have been designed to prove entailment or answer queries over a set of statements expressed in first-order logic or its fragments. Despite advances in automated theory proving (ATP), first-order logic reasoning is not widely used in practice in part due to difficulties with reasoning with very large theories.

Several researchers have explored whether machine learning can be used to improve the performance of first-order reasoning, similar to how AlphaGo used deep learning to improve AI game playing (Silver et al. 2016). There have been some promising results, but to date, no significant progress. We hypothesize that best way to make progress on the problem is solve three subproblems: representation, learning strategy and control strategy. To make the problem more amenable to study, we have restricted our initial investigation to the Horn fragment of first-order logic.

This work contributes to a general neurosymbolic approach for logical reasoning that has three key components: 1) an embedding model that maps logical statements to vectors, 2) a scoring model that represents the likelihood that a path leads to a successful answer, and 3) a *guided* reasoner, where the choices of a backward-chaining reasoner are scored by the neural model. Prior approaches focused on improving the scoring model and guided reasoner; here we attempt different training methods for the embedding model

to learn more useful embeddings and increase the overall performance.

In particular, our contributions are

- Three improvements to the process for learning an embedding model for logical statements
- An approach to evaluate the quality of the embeddings to capture their intended semantics
- An evaluation of the embeddings ability to improve the efficiency of reasoning using a downstream scoring model

## Background

Neurosymbolic AI seeks "to integrate neural network-based methods with symbolic knowledge-based approaches" (Sheth, Roy, and Gaur 2023). This can include a broad range of topics from generating embeddings of knowledge graphs to training a neural network to predict whether one logical statement entails another. One of the earliest attempts to combine logic rules and neural networks was KBANN (Towell and Shavlik 1994). KBANN takes propositional Horn rules and directly encodes them into the neural network. Kijsirikul and Lerdlamnaochai (2016) train a neural network that can perform inductive learning on first-order logic data. However, their architecture only allows the input of data representing a conjunction of atoms, and the output is a set of classes. There is no way to incorporate axioms into their reasoning. Rocktäschel and Riedel (2017) trained a neural network to perform unification and apply a backward-chaining-like process. This network was used to predict missing atoms in a KB.

Recently, there have been reasoners that attempt to leverage LLMs. AlphaGeometry generates proofs to solve Olympiad geometric problems (Trinh et al. 2024). It uses forward inference to find conclusions from a starting premise, then uses a language model to generate auxiliary points and retries the forward inference if the current proof space fails. Rather than being query-driven, AlphaGeometry deduces new statements exhaustively. ReProver, another LLM-based theorem prover for LeanDojo selects premises from large math libraries (Yang et al. 2023). ReProver scores the retrieved premises using an LLM. Inherent in using these approaches is having unlimited access to an LLM and the ability to fine-tune it to the needs of reasoning.

Other researchers have used various representations for logical statements, particularly in the context of automated theorem proving. Jakubův and Urban use an approach based on term walks of length three (Jakubův and Urban 2017). They parse each logical statement and create a digraph of it. They then extract every sequence of three nodes from this graph and create a code for it. This code has one dimension for each possible sequence; thus for a vocabulary $\Sigma$, the vector must have $|\Sigma|^3$ dimensions. This approach does not scale to KBs that have many constants.

Crouse et al. (2021) propose a chain-based approach that starts with a graph like Jakubův and Urban, but extracts patterns from a clause that start with predicates and end with variables or constants. Each sequence is hashed using MD5 and then that value is further reduced to $d$ dimensions by the modulo operator. Negative clauses are represented by concatenation of an additional $d$ dimensions. When creating the patterns, all variables are replaced by the symbol "*". This models the semantics that a variable can match with any term, but not that if the same variable appears multiple times in an expression, it must match the same term in every occurrence. Furthermore, the hashing and modulo operator addresses the scalability problem of termwalk, but does mean that statements are placed in latent space at random, as opposed to based on some inherent notion of similarity.

Arnold and Heflin (2022) proposed that embeddings for logical atoms could be learned in a way that retains the semantics of variables. A core operation used by any first-order logic reasoning algorithm is unification, where $\text{Unify}(\alpha, \beta)$ returns a substitution $\sigma$ such that $\alpha\sigma = \beta\sigma$ or fails. A set of unifying atoms and another set of non-unifying atoms can be found procedurally. Using a neural network that optimizes triplet loss (Vassileios Balntas and Mikolajczyk 2016), these atoms can be mapped into embeddings such that unifying atoms are near, and non-unifying atoms are far away. We describe Arnold and Heflin's approach and our improvements in subsequent sections.

## Learning to Improve Logical Inference

Backward-chaining is an algorithm for Horn logic inference that operates by starting with a goal and systematically working backward through a series of rules and known facts to determine the conditions required to achieve the goal. Traditional backward chaining reasoners often rely on a brute-force approach to explore potential solutions, which can lead to inefficiencies as the complexity and scale of the problem increases. Even relatively small knowledge bases of thousands of statements can result in searches of millions of nodes unless they have been carefully designed by a knowledge engineer. Previous work looked at learning a scoring model to direct the search along promising paths and compared chainbased, termwalk, and unification as a means for solving this problem (Jia et al. 2023). Our work improves on the previous unification approach to learn meaningful embeddings for logical statements.

The workflow of our system is shown in Figure 1. Starting with a knowledge base (KB) of facts and rules, we use a forward-chaining reasoner to infer new facts from the exist-

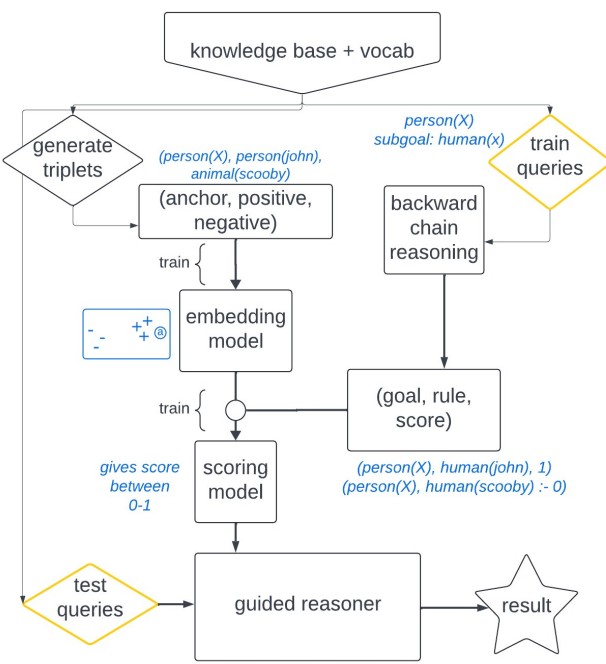

Figure 1: Representation of model structure

ing information. From these new facts, we randomly substitute constants with variables and divide the resulting list into one hundred training queries and one hundred test queries. We then solve the training queries using a randomized backward chaining reasoner, exploring all possible paths to a solution within a predefined depth limit. For each path, we assign $< goal, rule, score >$ tuples to the results. Here, $goal$ refers to the target query, $rule$ denotes the logical statement(s) used in the proof, and $score$ represents the effectiveness of a rule, with a score of 1 indicating a successful rule and 0 indicating a rule that does not lead to a solution.

We have a two-step training process. First, we learn embeddings for atoms using triplet loss (Vassileios Balntas and Mikolajczyk 2016). Triplet loss requires a set of $\langle anchor, positive, negative \rangle$ tuples and learns embeddings that place the $anchors$ close to the $positive$ examples, and far from $negative$ examples. Arnold and Heflin's original approach for training the embedding model (Arnold and Heflin 2022) involved generating a list of atoms at random, and for each atom, selecting from the same list a unifying atom to serve as the positive example and an atom that does not unify to serve as the negative example. Together, the anchor, positive, and negative atom are mapped onto a 50-dimensional embedding space using a three-layer network that minimizes triplet loss.

Second, the goal/rule/score tuples are converted into vectors using the embedding model. Finally, supervised learning with a two-layer neural net is used to train a scoring model. Initial experiments have show that the learned model is often able to significantly reduce backward-chaining search, sometimes by an order or magnitude or more. How-

| Repeat Chance | Repeat Const. | Repeat Var. | Atoms |
|:---:|:---:|:---:|:---:|
| 0% | 85.5% | 14.5% | 110 |
| 15% | 58.8% | 41.2% | 476 |
| 30% | 54.5% | 45.5% | 864 |

Table 1: Repeating terms in anchors. *Repeat Chance* is the additional probability that a anchor repeats a term. *Repeat Const. and Repeat Var.* represent the % of produced RTAs that repeat constants and variables respectively. *Atoms* is the total number of RTAs from 10,000 generated atoms.

ever, for some queries (and even some KBs) the improvements have failed to materialize.

## Our Approach

In this paper, we experimented with techniques to improve the quality of the embedding model, with the goal of improving performance in downstream tasks. We evaluated three types of improvements: First, we increased the likelihood to generate more atoms with repeated terms. Second, we define specific mutations for the atom to generate optimal training data, and lastly, we periodically train the model on samples with the highest loss (Harwood et al. 2017).

### Generating Atoms with Repeated Terms

We define repeated term atoms (RTAs) as logical facts that are not produced frequently by uniform distribution, but have additional semantics that the embedding model should learn. For example, $loves(X, X)$ will unify with far fewer atoms than $loves(X, Y)$. Initially with our uniform random generation, the likelihood that we had a second term repeat was $1/4(1/v + 1/c)$, a computed probability of 2.6% when the number of variables ($v$) was 10 and the number of constants ($c$) was 200. We modified the anchor generation process to produce a repeated term with a fixed probability. If the random atom has an arity $\geq 2$ then the *repeat chance* determines the likelihood that the next term will be a repeat of a preceding one. Table 1 shows the impact of setting the term repeat chance to different values. The prior embedding approach only generated 110 (out of $10,000$) atoms with a repeated constant or variable. By creating an additional repeat chance of $15\%$, we are able to produce $3.3\times$ as many atoms with a repeated term. Note, since this process only applies to atoms with arity $\geq 2$, the resulting repeated term atoms is less than $15\%$ of the total anchors. As expected, increasing the repeat chance to $30\%$ nearly doubles the atoms with repeated terms. However, our preliminary experiments showed that generating too many RTAs resulted in a reduction in downstream performance. In our following experiments we use a repeat chance of $15\%$ that has improved the overall performance of our embedding model and guided reasoner. In an ablation study discussed later, we measure it's effectiveness on synthetic knowledge bases.

### Mutating Atoms to Generate Triplets

An important observation we made is that the conditions for defining positive and negative atoms are already

---

**Algorithm 1: Anchor Mutation**

**Input**: An anchor atom $anchor$
**Output**: A mutated triplet

1: $positive \leftarrow anchor$
2: $negative \leftarrow anchor$
3: # *Modify positive atom "$p(\alpha_1, \alpha_2, ...\alpha_i)$" arguments*
4: **for** $\alpha_i$ in $positive$ **do**
5:     **if** RAND() $\geq 0.5$ **then**
6:         **if** $\alpha_i$ is a Variable **then**
7:             # *with uniform chance*
8:             $sub \leftarrow \{\alpha_i/C_{new}\}$or$\{\alpha_i/V_{new}\}$
9:         **else**
10:             $sub \leftarrow \{\alpha_i/V_{new}\}$
11:         **end if**
12:         apply substitution $sub$ to $positive$
13:     **end if**
14: **end for**
15: # *Modify negative atom "$n(\alpha_1, \alpha_2, ...\alpha_i)$" arguments*
16: **for** $\alpha_i$ in $negative$ **do**
17:     **if** RAND() $\geq 0.5$ **then**
18:         **if** $\alpha_i$ is a Constant **then**
19:             $\alpha_i \leftarrow C_{new}$
20:         **end if**
21:     **end if**
22: **end for**
23: **if** UNIFY($anchor$, $negative$) **then**
24:     $n \leftarrow P_{new}$
25: **end if**
26: **return** $\langle anchor, positive, negative \rangle$

---

well defined. For example, given an anchor atom like $mom(X, john)$, we can convert this into a positive or equivalent atom by replacing $X$ with a constant, such as $mary$. To make it a negative atom, we can replace $john$ with a different constant, such as $jill$. Generally, to derive a positive or equivalent logical statement from an anchor, we can change a variable into another variable or constant, or vice versa. On the other hand, to obtain a negative or non-equivalent statement, we can either replace a constant with another constant or alter the predicate itself, such as changing mom to a different predicate.

Defining these rules allows us to change the original generation of triplets into something much more effective, and further allows us to control the data our model receives for training. We generate a list of anchor atoms as before, but instead of *finding* the positive and negative atom in a predefined list, we simply generate them by *mutating* the properties of the anchor atom using a controlled, random approach.

The pseudo-code of the mutation approach is displayed in Algorithm 1. For each anchor atom, the code creates a positive (unifying) and negative (non-unifying) atom. To create a positive example, the code mutates each term $\alpha_i$ with a 50% probability. If it is a variable, it can be mutated to either a constant or a new variable. If it is a constant, then it must be mutated to a variable (since a constant can never unify with a different constant). These mutations are treated as substi-

tutions that are applied to the entire atom, so that repeated occurences will be replaced with the same new term (line 14). To create a negative example, the code mutates each term $\alpha_i$ with a 50% probability. In the case where the final negative atom still unifies with the anchor (e.g., if it only has variable terms), the predicate is replaced with a new predicate. As written, the pseudo-code only produces one triplet per anchor, but it can be called several times per anchor to produce more triplets.

The initial randomized triplet generation approach typically results in many positive, negative, and anchor triplets that inherently repeat since they can be rearranged for equivalence. An example case being the triplet $< female(Y), mother(mary), father(john) >$ which can be rearranged as $< mother(mary), female(Y), father(john) >$. It's important to observe that our use of Triplet Loss reduces the distance between the anchor and positive atom, while increasing the distance between the anchor and negative atom. When rearranged for equivalence, the anchor-positive pair retains the same distance-closing behavior while the anchor-negative pair separates $< mother(mary), father(john) >$ instead of $< female(Y), father(john) >$. It can be argued that the model gains *some* knowledge from rearranged triplets, but our research shows that it's more valuable for the model to learn on a completely new set of triplets. By preventing cases like these and multiples of the same triplet, our model can train on more diverse and valuable information. With our mutation approach, we can easily limit and control the amount of repeated anchors and enrich the training data while still keeping it organic. This adjustment captures rare but potentially significant semantics that could improve the ability of the model to handle unseen queries.

The number of possible atoms depends on the vocabulary of the KB. We describe the vocabulary for $kb$ using $np_{kb}$, $nc_{kb}$, $nv_{kb}$ and $ma_{kb}$ to represent the number of predicates, number of constants, number of variables and maximum arity. Then the maximum number of unique atoms is $np_{kb} * (nc_{kb} + nv_{kb})^{ma_{kb}}$. As this quantity grows, more data will be needed for the model to capture the semantics of the atoms. An important consideration is how many triplets should contain the same anchor. Our experiments have suggested that if this number is too low there is insufficient data to learn how that anchor relates to other atoms, but if the number is too high, then there may be too few anchors to properly generalize to unseen anchors. We define the triplets related to a single, unique anchor as the "triplets per anchor" (TPA). Through the later discussed ablation study (see Table 4), we observed that increases in the number of TPA often lead to increases in performance of the model, largely depending on the size of the knowledge base and vocabulary used. We define a target number of triplets to generate synthetically, and based on a desired number of TPA, the number of unique atoms changes. For instance, if our target for training is 500k triplets, and our desired number of TPA is 50, the embedding model will train on 10k unique anchors, with each anchor having 50 related triplets. Similarly our training target could remain 500k triplets, but a TPA of 20 would generate 25k unique anchors. In many ways, this relationship can be observed as a trade-off function between the amount of unique anchors generated and the number of triplets generated per anchor.

## Training on Hard Samples

The final improvement to the embedding model involves a technique that focuses on training with the hardest samples. In prior work our training loss would tend to flatten out early, so we focus training on semi-hard and hard samples, which improves the generalization of the model (Harwood et al. 2017). Similar improvements from training on hard samples have been observed in Convolutional Neural Networks through the work of Sahayam, Zakkam, and Jayarama (2023). After generating a synthetic dataset of a few hundred thousand triplets, we continuously use half of the set with the highest loss to represent our "hardest" samples. During training at every n-epochs, we validate the model's performance and continue refining it using a subset of the original generation of triplets, focusing on the training samples with the highest loss. This forces the model to learn and focus on difficult triplets that will help it solve queries quickly. As we train the embedding model on the hardest samples, the "hard" samples gradually become "easy" samples, and the previously difficult samples are removed from the new subset of hard samples collected every 10 epochs from the original synthetic dataset. Through this practice our model is still successful in training on all synthetic triplets generated.

# Evaluation

In this section we evaluate the proposed changes to the training of the embedding model. First, we examine the extent to which the embedding model achieves it goal. We then conduct a study to see how the changes impact the performance of the end-to-end system. Finally, we conduct an ablation study to determine the extent to which each change contributes to performance.

## Semantics of our Embeddings

The goal of our embedding model is to capture fundamental semantics about first-order logic, and we train it to place unifying atoms close and non-unifying atom far away. In this section we compare Arnold and Heflin's prior embedding training approach to one that uses the ideas proposed in this paper. Unlike Arnold and Heflin, we determine how well the embeddings generalize by creating a test set of anchors that is distinct from the training anchors, and then generating positive and negative examples for these anchors.

In Figure 2, we present a histogram of the accuracy of the new embedding model to determine how well the model associates 50,000 unseen anchors with a unifying example (anchor, positive pair), and how well it associates a non-unifying example (anchor, negative pair). In this figure, there are two peaks for the unifying examples. We hypothesize that first peak is because some positive examples likely had "perfect" predictions because some anchors in the validation set were positive examples in the training set. Our histograms are different than those originally presented by

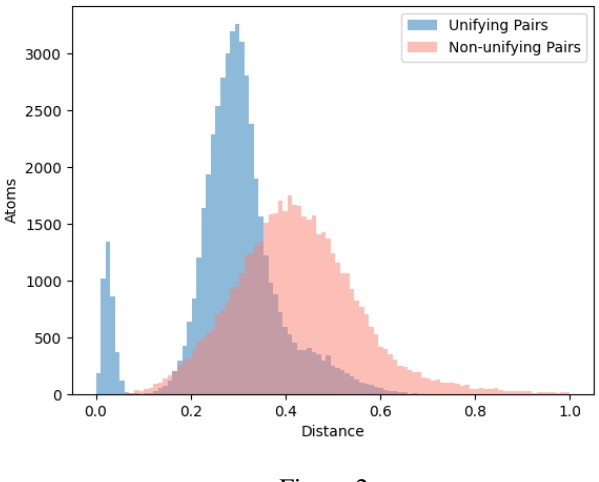

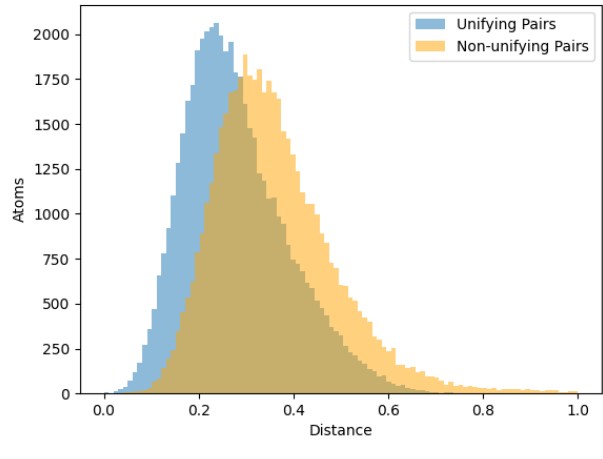

| Figure 2 | Figure 3 |

Histogram of similarity scores for unseen atoms and their unifying pairs (blue) and their non-unifying pairs (red & orange) on a KB using 20 predicates, 200 constants, 10 variables, and a maximum arity of 2. Figure 2 represents the new embeddings approach, and Figure 3 represents the prior approach.

Arnold and Heflin since we measure training loss with pairwise distance as opposed to cosine distance (Arnold and Heflin 2022). Pairs closer to 0 in latent space indicate a very similar positive example, while pairs farther away indicate an anchor may not be related to it's pairing. In Figure 3 we contrast the results of our new embeddings approach with that of Arnold and Heflin (2022) by creating a histogram under the same conditions. Note, because the different measures result in differences in scales, we normalize distances in both diagrams to the range [0,1] before displaying in the histograms. When comparing the histograms, one can see that there is less separation between the two classes of data in the prior approach.

We can also observe statistical differences with Total Variation (TV) distance:

$$D_{TV}(P,Q) = \frac{1}{2}\sum_{x \in X} |P(x) - Q(x)| \qquad (1)$$

TV distance measures the maximum difference between two discrete probability distributions by comparing the frequencies of the similarity scores. It ranges from 0 (identical distributions) to 1 (completely unlike distributions). From recent prior work in Figure 3 we compute a TV distance of 0.28, and from our improvements to the embedding model in Figure 2, a TV distance of 0.52. Generally, we have seen that the more TV distance between positive and negative similarity distributions, the less nodes our model explores in the search space until it reaches a solution. This metric hints at the significance of our improvements over prior work, because the embedding model is able to properly distinguish between a related and non-related atom.

| Reasoner | Size | Mean Nodes | Median |
|---|---|---|---|
| Standard | 250 | 17,204.0 | 1998.7 |
| Previous Embeddings | 250 | 981.8 | 3.4 |
| New Embeddings | 250 | 42.5 | 3.2 |
| Standard | 375 | 167,297.8 | 3035.3 |
| Previous Embeddings | 375 | 129,393.9 | 11.0 |
| New Embeddings | 375 | 33,280.0 | 2.8 |
| Standard | 500 | 3,419,493.6 | 552,639.1 |
| Previous Embeddings | 500 | 8,481,922.1 | 2.8 |
| New Embeddings | 500 | 3,916,035.3 | 2.8 |

Table 2: Comparing reasoners on different KB sizes. *Mean Nodes* is an average taken across 5 different KBs generated for the size. *Median* is an average of the median nodes explored in each KB.

## Reasoning Performance

To test how our proposed embedding approach impacts downstream tasks, we consider three different KB sizes: 250 statements, 375 statements and 500 statements. For each size, we generate a set of 5 synthetic KBs, as performance can be very different between KBs of the same size. We use 200, 300 and 400 constants in the 250, 375 and 500 statement KBs. Since larger vocabularies should require more training, we trained 250KB using 100k triplets, and both the 375KB and 500KB were trained using 200k triplets.

We have three representative reasoners, 1) a standard backward-chaining reasoner (*Standard*), 2) a guided reasoner trained using the original unification embedding approach (*Previous Embeddings*), and 3) a guided reasoner trained using everything proposed in this paper (repeated terms, mutated atoms, hard samples) to improve embeddings (*New Embeddings*). Both the previous embeddings and new

| Reasoner | Nodes explored | Time |
|---|---:|---|
| Baseline | 981.8 | 39.4 |
| Hard samples | 325.2 | 40.2 |
| Mutations | 231.3 | 25.0 |
| Repeated Terms | 99.0 | 24.4 |
| All improvements | 42.5 | 19.2 |

Table 3: Ablation study results. *Nodes explored* is an average taken across 5 different KBs. *Time* is an average measured in seconds.

embeddings systems use the Min Goal control strategy of Schack et al. (2024).

We report our results in Table 2. For each of the 5 KBs, we generated 100 unique queries. We collected the mean nodes explored across the 500 queries (100 per KB) to obtain our *Mean Nodes* metric displayed in the table. To obtain the *Median* nodes metric, we averaged the median nodes explored across each of the 5 KBs. The rows for *Standard* and *Previous Embeddings* are results reported in Schack et al. (2024). The *New Embeddings* experiments were conducted under (almost) identical conditions. The only difference is in the maximum number of nodes before a query **fails**: 100,000,000 in Schack et al., while 10,000,000 for the *New Embeddings* reasoner. This adjustment allowed the experiments to be conducted in days as opposed to weeks. To ensure the results are comparable, we adjusted the nodes for every failed query from 10,000,000 to 100,000,000. Some of the failed queries might have actually terminated with fewer nodes if given the larger cutoff, which means the *New Embeddings* approach could be even better than reported here.

We make the following observations from the data. Across each size, the medians are smaller than the mean because of a few large outliers present in each knowledge base. The medians for the two embedding approaches are orders of magnitude smaller than those of the standard reasoner. With the exception of the 375 KBs, the previous embeddings and new embeddings have very similar medians. The results for the means show that the new embeddings are much better than those of the previous system, ranging between 9.2% and 46.2% of the mean nodes explored. This basically means there are fewer large outliers, possibly indicating that the system generalizes better. It also scales to larger KBs better, although it can still be impacted by the occasional outlier. For example, we had an unexpected outcome where one set of queries in the 500KB *New Embeddings* reasoner raised the mean nodes explored significantly, and believe that this is due to the stochastic nature of training since all other individual experiments resulted in a much lower nodes explored.

## Ablation Study

To understand how each of our proposed embedding modifications impacted performance, we conducted an ablation study. Each ablation was conducted with 5 different knowledge bases of size 250. We used 20 predicates with a maximum arity of 2, and 200 unique constants when generating each synthetic KB. To keep the environment of our experi-

ments as constant as possible, we also utilize the same KB and testing queries across the baseline and each ablation. Our results are shown in Table 3. *Baseline* represents recent prior work on learned embeddings (Schack et al. 2024), *Mutations* represents the triplets per anchor approach, *Repeated terms* represents the increase in repeated term atom generation, and *Hard samples* represents our minimization of examples with the highest triplet loss.

We first focus on measuring improvements from our specific mutations and triplets per anchor approach. The result is a mean nodes explored of 231.3, showing a 76.4% decrease from prior work. This decrease represents a reduction in time and resources needed to answer a set of queries. We hypothesize that this approach which also controls the number of triplet repeats per anchor is another parameter that could be fine-tuned to reduce the nodes explored. The relationship, which we previously referred to as the trade-off function, implies that generating a specific number of triplet repeats per anchor is better than giving our embedding model more data to train on, and that this relationship varies based on the number of unique predicates and constants present in our knowledge base.

We investigate this trade-off function and the effectiveness of the mutation technique, particularly on a KB of size 500 with 20 unique predicates and 400 unique constants. In Table 4, we report our results when using a fixed number of 200,000 training triplets. The number of triplets per anchor has a noticeable impact on performance. Increasing the synthetic data generated in many obvious cases increases training time, and strains computational resources like memory. With our work, we achieved a 49.9% average decrease in nodes explored from 5 to 20 TPA without increasing the amount of synthetic training data generated. However, when the TPA is too large (30 in this experiment), end-to-end performance begins to suffer. We hypothesize that this is due to the model not seeing enough unique anchors to generalize well.

Next, we observe the improvements of an increase in repeated terms by no longer generating anchor atoms uniformly. The result is a mean nodes explored of 99.0, representing a 89.9% decrease from prior work. This is another parameter that could be fine-tuned to improve overall performance, but we are unable to identify a relationship for the ideal number of RTAs needed to achieve lower nodes explored, nonetheless, a small increase in RTAs tends to improve our results significantly.

Finally, we examine the effects of training our model on

| TPA | Nodes explored | Median |
|---|---|---|
| 5 | 7,816,589.1 | 2.8 |
| 10 | 5,239,567.7 | 2.8 |
| 20 | 3,916,035.3 | 2.8 |
| 30 | 8,638,852.5 | 2.8 |

Table 4: Results from experiments using 5 KBs with 500 statements. *TPA* referring to the number of triplets related to an anchor. *Nodes explored* is an average across 5 KBs and 100 queries.

the half of the original dataset with the highest triplet loss. We observed an average of 325.2 nodes explored, a notable 66.8% decrease. Although "hard examples" lags slightly in terms of improvements against the other ablations, it still shows promise. In future work we will improve on this technique by training on the most valuable samples with triplet mining, as opposed to training on the hardest half of examples (Harwood et al. 2017).

In summary, our ablation study shows that individually each of our three proposals significantly improves upon Arnold and Heflin's original approach for learning embeddings. Of these, the training for repeated terms has the greatest impact, followed by generating positive and negative examples by mutation, and then training using hard samples.

## Conclusion

Our team's work is important to the efforts of researchers to reduce the computational resources required to infer solutions to logical queries and mathematical proofs. By focusing on approaches to train an embedding model, we have showed that while working on methods for scoring and choosing queries are important, there are downstream benefits by improving the embeddings for logical inference. The choice of representation for symbolic atoms and the process for learning these representations can have a significant effect. To put it pithily, "representation matters." Starting from a goal that the right representation for logical atoms is one that places unifying atoms close in latent space, we have demonstrated approaches to help us learn good representations: intentionally oversample anchor atoms that have repeated terms, create positive and negative samples by mutating these anchors, and use a training process that regularly emphasizes the hard atoms.

Through this work, our observations led us to hypothesize that more relationships likely exist, like the previously mentioned trade-off function where using more triplets per anchor becomes better than adding more data. We also hypothesize an additional relationship between the size of our knowledge base vocabulary, and the trade off function. In any case, there are many parameters we could have fine-tuned to optimize performance, and we plan to investigate the relationship between these parameters. Future work will test larger and more realistic knowledge bases, which often tend to be less complex than our synthetically generated KBs, with our main objective to demonstrate the impact of our improvements on real-world systems. Once we have a good understanding of how to design embeddings for Datalog, we would like to extend the approach to embeddings for full first-order logic, and evaluate whether similar benefits can be achieved for the resolution algorithm.

## Acknowledgments

This work was conducted as part of an REU site supported by the National Science Foundation under Grant No. CNS-2051037.

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
