# OpenReview forum: "An Evaluation of Approaches to Train Embeddings for Logical Inference"
_AAAI.org/2025/Workshop/NeurMAD — AAAI 2025 Workshop NeurMAD Submission_

### Official Review · Reviewer_gmc8 · 2024-12-20
**Interesting paper that is quite relevant to the workshop**

**Rating:** 7
**Confidence:** 5

**Review:**

The paper considers the problem of neurosymbolic learning and aims to improve it by improving the training of the embedding model for logical statements, evaluating the quality of the embeddings to determine if they model the original semantics and finally improving the reasoning using a downstream scoring function.

The paper is well written and is quite relevant to the topic of the workshop. I would certainly like to see this presented at the workshop and get some more detailed feedback from the poster session.

Of course, standard improvements including improving the empirical and theoretical analyses, adding different types of KGs and baselines are possible which I am sure that the authors would consider before submitting this to a conference.

Over all, this is quite exciting work that I enjoyed reading.

---

### Official Review · Reviewer_pGaA · 2024-12-26
**Good task**

**Rating:** 7
**Confidence:** 3

**Review:**

This paper describes a framework to train a neural embedding model for logical inference tasks. The core idea is through 'anchor mutation': triplets such (anchor, positive, negative) are generated via mutation, and an embedding model learns to embed unifying atoms (anchor and positive) closer (on the embedding space), and non-unifying atoms (anchor and negative) far away. This is closer to the idea of contrastive learning to me, which makes a lot of sense.

Strength:
- well-motived task
- clearly presented algorithm

Weakness:
- I would love to see a more formal definition of repeated term atoms (RTAs) and perhaps a few more words on its effect on the downstream tasks.
- More explanations on Table 2. How is the new embedding strategy different from the previous embeddings? Would it be correct to say that over large KBs the new embeddings only improve upon the mean nodes metric while stay the same upon the median metric? Would it be possible to perform worse over even larger KBs (on the medium nodes explored metric)?

---

### Decision · Program_Chairs · 2024-12-30

**Decision:**

Accept

**Comment:**

 This is a well-written paper and also suitable for this workshop. We agree with the opinions of the reviewers.